# Dreaming about Dogs: An Online Survey

**DOI:** 10.3390/ani10101915

**Published:** 2020-10-19

**Authors:** Michael Schredl, Christian Bailer, Muriel Sophie Weigel, Melina Sandra Welt

**Affiliations:** Central Institute of Mental Health, Medical Faculty Mannheim, Heidelberg University, Postfach 12 21 20, 68072 Mannheim, Germany; cbailer@mail.uni-mannheim.de (C.B.); muweigel@mail.uni-mannheim.de (M.S.W.); melina.welt@web.de (M.S.W.)

**Keywords:** dog dreams, threatening dreams, dog owner, dream recall frequency

## Abstract

**Simple Summary:**

The findings of the survey indicate that waking-life experiences with dogs (owning a dog or negative experiences with dogs in the past) affect dreaming in a significant way. On the one hand, dog owners dream about dogs more often and had overall positively toned dreams, whereas persons with negative experiences with dogs in their waking life reported a higher percentage of dreams with threatening dogs.

**Abstract:**

Dogs have been close human companions for millennia and one would expect—according to the continuity hypothesis of dreaming—that dogs are also quite common in dreams. Previous studies showed that the percentages of dreams that include dogs range from about 1.5% to 5%, but studies relating waking-life experiences with dogs with dreams about dogs have not been carried out. In total, 1695 persons (960 women, 735 men) completed an online survey that included questions about dreams and waking-life experiences that included dogs. The findings indicate that dogs show up, on average, in about 5% of remembered dreams, but this percentage is much higher in the dreams of dog owners and persons with close contacts with dogs. Moreover, the active time spent with a dog and the proximity during sleep is also related to a higher percentage of dreams that include dogs. Although dreams including dogs are on average more positively toned than dreams in general, about 11% of the dog dreams included threatening dogs. Persons who had negative experiences with dogs in their waking lives reported more threatening dog dreams. The results support the continuity hypothesis and it would be very interesting to conduct content analytic studies with dream samples obtained from dog owners to learn more about the variety of interactions between dreamers and dogs.

## 1. Introduction

Dogs have been human companions for millennia [1] and are widespread all over the world. In Germany, for example, dogs are present in 19% of all households, with a total number of about 9.4 million dogs [2]. Given the closeness of human–dog relationships, e.g., dogs as part of the family, dogs as best friends, treating dogs like persons, and daily interactions such as walking the dog, feeding, etc. [3], one would expect—according to the continuity hypothesis of dreaming [4]—that dogs would also be quite common in dreams. Indeed, dogs are often the most frequent animals within dreams; the percentage of dreams including dogs ranges from about 1.5% to 5% [5,6,7,8,9,10,11]. There is a long history of speculating about the meaning of dog dreams that dates back to the second century AD [12], including the ideas that animals in general, but especially dogs, represent the animal nature of humans [13,14], the impulsive self of the child [15], the parents [16], or sexual impulses in adolescence [17]. Artemidorus (second century ACE) differentiated between hunting dogs (representing the deeds of the dreamer), house dogs (representing family and belongings), and Malteser dogs in dreams that represent the most delightful and pleasant things in life [12]. Despite the large amount of literature on the symbolic interpretations of dogs in dreams, there is a lack of empirical research looking, for example, into the simple question as to whether dog owners dream about dogs more often than persons who do not own dogs. The finding that animal rights activists dream more often about animals in general, and also report more dog dreams (9.51%), would be in line with the continuity hypothesis, since one might assume that individuals investing considerable effort in protecting animals also have a great love for them. In a single case study [10], it was shown that the newly bought dog of the romantic partner of the dreamer increased the number of dog dreams significantly from 0.53% in the year 2006 (overall, 189 dreams were remembered in that year) to 3.93% in the year 2007 (280 dreams). This is the first hint that living close to a dog affects the number of dog dreams.

Several studies [7,18,19,20] found that most animal dreams are negatively toned, e.g., being bitten or being chased. Van de Castle [7] reported that animals were attacking the dreamer in about 30% of the dreams, but were companions or playmates in only about 15% of the animal dreams; however, if the dream animal interacted with the dreamer positively, it was most likely a dog. In the single case study based on 108 dog dreams (out of a series of 8400), 9 (8.33%) included negative interactions with dogs (being threatened, being bitten, fighting with the dog) whereas 30 (27.78%) included positive interactions between the dreamer and a dog (the dreamer cares for the dog, plays with the dog, or the dog helps the dreamer). This research indicates that dreams including dogs could be either positive or negative (in addition to neutral dreams). According to the continuity hypothesis of dreaming [4], one would expect that individuals who have had negative experiences with dogs in their waking life would also have more negatively toned dog dreams.

The aim of this study was to investigate whether or not waking-life experiences with dogs affect dreams. We hypothesized that dog owners dream about dogs more often when compared to persons who do not own a dog. Moreover, it was expected that persons with negative dog experiences in their past would also report a higher percentage of negatively toned dog dreams.

## 2. Method

### 2.1. Participants

Overall, 1695 persons (960 women, 735 men) completed an online survey between 13 April 2020 and 20 April 2020. The mean age of the sample was 53.84 ± 13.99 years (range: 20 to 96 years). Concerning educational level, 0.53% had no degree, 12.80% had 9 years of schooling, 39.62% had O-levels (middle degree, “Realschule”, about 10 years), 23.13% A-levels (“Abitur”), 30.91% obtained a University degree, and 3.01% had doctoral degrees.

### 2.2. Research Instrument

For eliciting dream recall frequency, a 7-point scale (coded as 0 = never, 1 = less than once a month, 2 = about once a month, 3 = about two to three times a month, 4 = about once a week, 5 = several times a week, 6 = almost every morning) was presented. The exact wording was [21] “How often have you recalled your dreams recently (in the past several months)?” The retest reliability of this scale is high—r = 0.85 for an average interval of about 55 days [22]. The overall emotional tone of the dreams was measured via five categories (−2 = very negative, −1 = somewhat negative, 0 = neutral, +1 = somewhat positive, and +2 = very positive).

Next, the participants were asked to estimate the percentage of dreams that included dogs; the following explanation was included: “The dog does not have to be the focus of the dream. It can be your own dog but also unfamiliar dogs. If you have never dreamed of dogs, please enter a zero.” The emotional tone of the dreams that included dogs was—like the general emotional tone of dreams—measured on a five-point scale ranging from −2 to +2. The next question was: “What is the percentage of remembered dog dreams where the dog is threatening?”

The next section included items about dogs in waking life, starting with the question: “Are you or a household member currently the owner of one or more dogs?” There were four answering options: “Yes, I myself own a dog/dogs”, “Yes, there is a dog/dogs in my household”, “No, but I have owned a dog or lived with a dog in my household”, and “No, I have never lived with a dog.” The time spent with the dog was measured as follows: “How much time during a week do you actively spend with your own and/or another dog? Note: Active time includes, for example, playing, cuddling, petting, going for a walk, feeding, etc.” The typical sleep locations of the dog(s) were also elicited—within the bed, within the bedroom but outside the bed, within the household but outside the bedroom, and outside the household. Lastly, the participants were asked whether they had had negative/threatening experiences with dogs in the past. If so, the situation should be described briefly.

### 2.3. Procedure

Within the wisopanel online panel, persons with an interest in online studies and with heterogenic demographic backgrounds are registered. At the time of the study, 14,277 individuals were in the database. Almost all the participants live in Germany with less than 5% living in Austria and Switzerland. The survey was conducted in German. All registered persons received an email with the link to the study entitled “Everyday life and dreams”. The participation was voluntary and unpaid. The study was carried out following the rules of the Declaration of Helsinki of 1975, revised in 2013. In Germany, this type of study asking healthy human volunteers to participate in an online survey does not require ethical approval. This was confirmed by the Ethics committee of the University of Mannheim.

Statistical procedures were carried out with the SAS 9.4 software package for Windows. As the percentage variables (percentage of dog dreams, percentage of dog dreams with threatening dogs) were not normally distributed, they were categorized (see the results section below). The categorized variables were treated as ordinal variables. The reports about negative experiences with dogs in the past were classified into several groups: “Being bitten”, “Feeling threatening”, “Being attacked without being bitten”, “Loud barking”, “Another person or the own dog was bitten”, and “Others”.

Ordinal regressions (cumulative logit model) were used for analyzing the effects of waking-life variables, e.g., owning a dog, having owned a dog, time spent with the dog, or negative experiences in the past with dogs, on dog dream variables (dog dream percentage, emotional tone of dog dreams) controlled for age, sex, education, and dream recall frequency. All variables were entered simultaneously. Effect sizes for each predictor were computed based on the Wald Chi-Square values obtained by the ordinal regression analyses and sample size according to the formula given by Cohen [23]. This option for computing effect sizes was chosen because (1) the dependent variable is ordinal and (2) the ordinal regression coefficient that is tested with the Wald Chi-Square is controlled for possible confounding effects, as the effects of all other variables entered simultaneously into the regression are particalled out.

## 3. Results

The distribution of the estimated dream recall frequency was as follows: never (8.69%), less than once a month (18.79%), about once a month (8.87%), about two to three times a month (13.36%), about once a week (18.62%), several times a week (22.99%), almost every morning (8.69%), and there were three missing values. The mean of the percentage of dreams including dogs was 5.52 ± 14.04% (*n* = 1695). The distribution of the categorized variables is depicted in Table 1. Overall, 525 participants (about 31%) reported that they had dreams that included dogs. The emotional tones of dreams that include dogs are shown in Table 2; the positive dreams clearly outweigh the negative ones (mean: 0.64 ± 1.06). Compared to the general emotional tone of the dreams (mean: 0.07 ± 0.81; see Table 2), the emotional tone of dreams including dogs was significantly more positive (Wilcoxon Signed Ranks test: z = 9.4, *p* < 0.0001, *n* = 497, effect size = 0.940).

227 participants owned a dog, whereas 49 participants stated that there is a dog within their household that they do not own. About 50% never owned a dog or lived with a dog, whereas 29% had owned or lived with a dog in the past (25 participants did not complete this item). The percentages of dreams that included dogs for the three groups (dog owners and persons with dog[s] in the household were collapsed into one group) differed considerably; see Table 3. The difference between the “dog owner/dog(s) within the household” group and the “Never had a dog or lived with a dog” group was large (effect size = 1.079) and significant (for significance levels, see Table 4). Similarly, participants who had a dog or lived with a dog reported a higher dog dream percentage (effect size = 0.537) than participants that had never lived with a dog. In addition, high dream recall was associated with higher percentages of dreams including dogs, and females reported slightly higher percentages of dog dreams then males (see Table 4). Dog owners and/or persons with a dog living in the household also reported that their dog dreams are more positive than those of persons who had never lived with a dog (effect size = 0.689; see Table 4). The same was true for persons who had owned a dog or lived with a dog (effect size = 0.482; see Table 4).

The average active time spent with the dog was, for the dog owners group, 20.28 ± 12.82 h per week (*n* = 198). Several answers (above 60 h per week) had to be excluded as these participants most likely confused overall time spent with the dog and active time. Regarding the dog’s sleeping place, 66 participants stated that the dog sleeps predominantly within the bed, within the bedroom but outside the bed (*n* = 71), within the household but outside the bedroom (*n* = 78), or outside the household (*n* = 11; one missing value). Within the dog owners group, the regression analysis indicated that the amount of active time spent with the dog is significantly associated with the percentage of dreams that include dogs (effect size = 0.430; see Table 5). Moreover, the closer the proximity of the dog’s sleeping place to the dog owner, the more likely (marginally significant, effect size = 0.260) it is that the owner dreams about the dog (see Table 5).

About one third of the participants who reported dreams including dogs stated that at least in some of these dreams the dog(s) were threatening (see Table 6). Only a small percentage of individuals reported that their dog dreams are predominantly threatening. The mean percentage of dog dreams that included threatening dogs was 11.14 ± 25.67% (*n* = 522). About 25% of the 1583 participants who responded to the question reported negative experiences in the past. Overall, 356 brief descriptions were provided in the study; 48.60% referred to being bitten by a dog, 37.64% included a threat such as being attacked or chased but not bitten, 6.46% included incidents in which another person or the dog of the participant was bitten, and 4.49% referred to experiences in which loud barking startled the participant. In total, 10 reports (2.01%) dealt with other topics like not being able to control a dog, being startled by a dog causing an injury, or dog feces (neighbor’s dog) on the porch. As expected, the individuals reporting negative experiences with dogs in the past reported a higher percentage of threatening dog dreams (effect size = 0.371; see Table 7). Interestingly, younger persons also reported a higher percentage of threatening dog dreams, as did males and persons with lower education levels (see Table 7). Similar to the percentage of dog dreams when compared to all dreams, dream recall frequency was associated with a higher percentage of threatening dogs within their dreams that included dogs.

## 4. Discussion

The findings indicate that, on average, dogs show up in about 5% of remembered dreams, but the percentage is much higher with dog owners or persons with close contacts with dogs. Moreover, active time spent with the dog and proximity during sleep are also related to a higher percentage of dog-including dreams. Although dreams including dogs are on average more positively toned than dreams in general, about 11% of the dog dreams included threatening dogs. Persons who had had negative experiences with dogs in their waking life reported more threatening dog dreams.

From a methodological viewpoint, it has to be noted that the present sample was self-selected and not representative, despite the large age range and varying educational levels. Comparing the dream recall frequency distribution to representative samples [21] clearly indicated that high recallers are overrepresented. This is of importance, as dream recall frequency was related to the percentage of dreams that included dogs, and thus it was necessary to include this possible confounder into the regression analyses. The positive relationship between dream recall frequency and the percentage of dog-including dreams in relation to all remembered dreams could reflect a methodological issue in the retrospective approach, since Schredl [24] was able to demonstrate that low dream recallers are less accurate in estimating different dream characteristics retrospectively compared to high recallers. On the other hand, one might speculate that other variables play a role; dream recall frequency, for example, is related to openness to experience [25], and it could be hypothesized that persons with high openness might also be more curious about dogs and dream more often about them. Thus, it would be interesting to study the relationship between dream recall frequency and the percentage of dreams including dogs in more detail in the future. On the other hand, there was no selection regarding dog owners, as the study was entitled “Everyday life and dreams”, and included other topics like work-related dreams and family-related dreams. The figure of 16.5% of participants owning a dog or living with a dog in the household is close to the representative percentage of 19% of German households with a dog [2]. In contrast to previous studies [7,9,20], the percentages of dog dreams were elicited retrospectively, but previous research [26,27] comparing content analytic data derived from diary dreams with this retrospective format of eliciting percentages of specific dream topics (in that case, sport dreams) demonstrated that the obtained figures are comparable. Moreover, the study was conducted during the Covid−19 lockdown in Germany, i.e., sleep behavior and dream recall might have changed due to this conditions. However, previous studies, e.g., Schredl et al. (2014) [21], carried out within the same panel (wisopanel), yielded a similar distribution of dream recall frequency, that is, markedly higher dream recall on average compared to representative samples.

The percentage of dreams including dogs of about 5% is in line with previous results [5,6,7,8,9,10,11], especially if one considers that the 5% figure is a slight overestimation due to many high dream recallers in the sample. Overall, the emotional tone of dog-including dreams was more positive when compared to the overall general tone of dreams, especially in dog owners. This shift towards positive dream emotions was also found for music dreams [28,29]; since music is a pleasant leisure time activity, this is in line with the continuity hypothesis. The positive emotional tone of dreams including dogs can be interpreted in a similar way. It would be very interesting to elicit explicit dream content, especially in samples of dog owners, about the variety of interactions between the dog and the dreamer.

The very high percentage of dog-including dreams (about 19%) in dog owners—comparable to figures of the romantic partner showing up in dreams [30,31,32,33]—clearly indicate that dogs are typically very close companions [3]. The closeness to the dog (amount of active time spent with the dog, dog sleeping in the bed) is also related to the percentage of dreams including dogs. All this is in line with the continuity hypothesis of dreaming [4]. Furthermore, persons who owned dogs or lived with dogs dream more often about dogs than persons who never owned a dog, indicating that not only do current experiences with dogs affect dreams, but past experiences do too (cf. [34]).

Despite the finding that dreams including dogs are on average more positively toned than dreams in general, in about 11% of the dog dreams the dog threatens the dreamer, and a similar figure has been reported by Schredl [10]. The higher percentage of aggression related to animals [7,18,19,20] might be related to the type of animal (i.e., dogs are not as threatening as other animals), but also may be related to age, as the previous samples were much younger (children, students) compared to the present sample and, in the present sample, younger persons reported threatening dog dreams more often. Interestingly, men also reported a higher percentage of threatening dog dreams—possibly reflecting the higher percentage of physical aggression found in men’s dreams when compared to women’s dreams [8,35,36]. About 25% reported negative experiences with dogs in waking life, clearly indicating that human-directed aggression in pet dogs is an important issue [37]. These persons reported higher percentages of threatening dog dreams, indicating that a negative experience in waking life, e.g., being bitten by a dog during childhood, can have lasting effects on dream life.

## 5. Conclusions

To summarize, the findings of the present study clearly indicate that waking life experiences with dogs (positive and negative) affect dreaming and, thus, focusing exclusively on symbolic interpretations of dogs in dreams might not be appropriate. It would be very interesting to conduct content analytic studies with dream samples obtained from dog owners to learn more about the variety of interactions between dreamers and dogs.

## Figures and Tables

**Table 1 animals-10-01915-t001:** Percentage of dreams that include dogs (*n* = 1695).

Category	Frequency	Percentage
more than 40%	70	4.13%
20.01% to 40%	66	3.89%
10.01% to 20%	70	4.13%
5.01% to 10%	103	6.08%
0.01% to 5%	216	12.74%
0%	1170	69.03%

**Table 2 animals-10-01915-t002:** Emotional tone of dreams that include dogs and the general emotional tone of dreams.

Category	Emotional Tone of Dreams Including Dogs (*n* = 519)		General Emotional Tone of Dreams (*n* = 1531)	
	*n*	Percent	*n*	Percent
Very positive (+2)	113	21.77%	36	2.35%
Somewhat positive (+1)	205	39.50%	431	28.15%
Neutral (0)	123	23.70%	689	45.00%
Somewhat negative (−1)	59	11.37%	351	22.93%
Very negative (−2)	19	3.66%	24	1.57%

**Table 3 animals-10-01915-t003:** Means and standard deviations for percentage of dreams that include dogs and the emotional tone of dreams that include dogs.

Group	*n* =	Percentage of Dreams Including Dogs	Emotional Tone of Dreams Including Dogs (*n* =)
Dog owners/Dog(s) within the household	276	19.19 ± 23.51%	0.95 ± 0.94 (204)
Owned a dog or lived with a dog in the past	483	5.68 ± 12.67%	0.71 ± 0.99 (188)
Never had a dog or lived with a dog	911	1.32 ± 6.04%	0.05 ± 1.10 (122)

Sample sizes for the emotional ton of do dreams are smaller as participants who had no dog dreams were omitted.

**Table 4 animals-10-01915-t004:** Ordinal regression analysis for the categorized (ordinal) percentage of dreams that include dogs and emotional tone of dreams that include dogs.

Variable	Percentage of Dreams Including Dogs (*n* = 1668)			Emotional Tone of Dreams Including Dogs (*n* = 513)		
	SE	χ^2^	*p*	SE	χ^2^	*p*
Age	−0.0506	2.4	0.1203	0.0720	2.5	0.1172
Gender	0.0799	5.9	0.0152	−0.0825	3.3	0.0696
Education	0.0059	0.0	0.8544	−0.0101	0.1	0.8223
Dream recall frequency	0.3258	85.7	<0.0001	0.0100	0.0	0.8272
Dog owners in the past vs. Never lived with a dog	0.3641	112.1	<0.0001	0.3079	28.2	<0.0001
Dog owners/household vs. Never lived with a dog	0.6208	376.1	<0.0001	0.4381	54.4	<0.0001

SE = Standardized estimates.

**Table 5 animals-10-01915-t005:** Ordinal regression analysis for the categorized (ordinal) percentage of dreams that include dogs (for dog owners only).

Variable	Percentage of Dreams Including Dogs (*n* = 198)		
	SE	χ^2^	*p*
Age	−0.0722	1.0	0.3150
Gender	0.0598	0.7	0.4058
Education	0.1338	3.5	0.0628
Dream recall frequency	0.2750	13.7	0.0002
Active time with the dog (h/wk.)	0.3641	9.0	0.0027
Sleep place of the dog (1 = outside to 4 = within the bed)	0.1361	3.3	0.0704

SE = Standardized estimates.

**Table 6 animals-10-01915-t006:** Percentage of dog dreams that include threatening dogs (*n* = 522).

Category	Frequency	Percentage
more than 50%	40	7.66%
10.01% to 50%	57	10.92%
0.01% to 10%	72	13.79%
0%	353	67.62%

**Table 7 animals-10-01915-t007:** Ordinal regression analysis for the categorized (ordinal) percentage of dreams that include threatening dogs.

Variable	Percentage of Dreams Including Threatening Dogs (*n* = 514)		
	SE	χ^2^	*p*
Age	−0.1915	12.3	0.0004
Gender	−0.1542	8.6	0.0034
Education	−0.1202	5.0	0.0250
Dream recall frequency	0.1802	10.0	0.0016
Negative experiences with dogs in the past (Yes/no)	0.2090	17.1	<0.0001

SE = Standardized estimates.

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
