# Peer review of "Dreaming about Dogs: An Online Survey"

_animals, 2020, doi:10.3390/ani10101915_

Round 1

Reviewer 1 Report

Dear authors,

I read the revised version of your paper. I only have a few residual comments:

Methods
-------
L124: recall frequency has never been defined, I suggest you define this term somewhere n Methods
L125-126: I still maintain that the Wald Chi-Squared test is used to test the significance of the effect, while the size (magnitude of the effect) can be obtained from, for instance, Cohen's d (or other metrics, depending on the type of variables). Please investigate

Author Response

Dear authors,

I read the revised version of your paper. I only have a few residual comments:

Methods
-------
L124: recall frequency has never been defined, I suggest you define this term somewhere n Methods

Answer: We added the exact text of the dream recall frequency item in the method section.

L125-126: I still maintain that the Wald Chi-Squared test is used to test the significance of the effect, while the size (magnitude of the effect) can be obtained from, for instance, Cohen's d (or other metrics, depending on the type of variables). Please investigate

Answer: I am not sure what the reviewer is aiming at. To my knowledge, there is no other way to obtain effect sizes for variables within an ordinal regression (the dependent variable is ordinal, i.e., often used metrics based on means and standard deviations do not work. We added the following explanation in the procedure section: "This option for computing effect sizes was chosen because (1) the dependent variable is ordinal and (2) the ordinal regression coefficient that is tested with the Wald Chi-Square is controlled for possible confounding effects as the effects of all other variables entered simultaneously into the regression are particalled out."

Reviewer 2 Report

The response to reviewers is fine and the paper has been modified as such.

Author Response

The response to reviewers is fine and the paper has been modified as such.

Answer: Thanks for the positive feedback.

This manuscript is a resubmission of an earlier submission. The following is a list of the peer review reports and author responses from that submission.

Round 1

Reviewer 1 Report

A dog was owned by 227 participants - should be: 227 participants owned a dog. (page4, directly after Table2)

The aim of the study was to investigate if waking-life experiences with dogs affect dreams. In this paper, a survey about how often people were dreaming about dogs, was performed. It showed in a nice way, the difference of how dogs appear in dreams, depending on the persons attitude towards dogs and their experiences with dogs. I think it was important and novel. It was an online survey, the questions were not only directed in the dog-direction, so I think the method was a good one. Even though, some could say, that it might be logical, that people with good experiences with dogs dream about them in a positive way and people who are afraid of dogs dream that way, it was not published till now. In my opinion the claims are convincing I am not an expert in this field but as I read in other papers, that surveys are a good way for this question, so material and methods are ok. Even though the present sample was self-selected and not representative, despite the large age range and varying educational levels. But this was discussed.

Author Response

Response to Reviewer 1

A dog was owned by 227 participants - should be: 227 participants owned a dog. (page4, directly after Table2)

Answer: Changed.

The aim of the study was to investigate if waking-life experiences with dogs affect dreams. In this paper, a survey about how often people were dreaming about dogs, was performed. It showed in a nice way, the difference of how dogs appear in dreams, depending on the persons attitude towards dogs and their experiences with dogs. I think it was important and novel. It was an online survey, the questions were not only directed in the dog-direction, so I think the method was a good one. Even though, some could say, that it might be logical, that people with good experiences with dogs dream about them in a positive way and people who are afraid of dogs dream that way, it was not published till now. In my opinion the claims are convincing I am not an expert in this field but as I read in other papers, that surveys are a good way for this question, so material and methods are ok. Even though the present sample was self-selected and not representative, despite the large age range and varying educational levels. But this was discussed.

Answer: Thanks for the positive feedback.

Reviewer 2 Report

Dear authors,

your paper "Dreaming about dogs: An online survey" is generally well written and deals with a rather exotic yet not devoid of interest topic.
The absence of line numbers made the revision work more difficult: you may consider adding line numbers to the revised version of the paper. More importantly, essential details must ne added to Methods to improve the understanding of your results and the reproducibility of the study.

Specific comments

Introduction
------------
- of dog dreams significantly from 0.53% in 2006 (189 dreams) to 3.93% (280 dreams): I think that "in 2006" is a typo

Methods
-------
- 2.1 Participants: where was the survey conducted?
- 2.3 Procedure: when you say ordinal regression, do you mean threshold models? (latent-variable model?) More details are needed on the model(s) used: please try to write the model explicitly, with the categorical (ordinal) variable to be analysed (response variable) and the independent variables used to model it. Otherwise it is not possible to fully understand your results and to try to reproduce your study
- 2.3 Procedure: it is not clear how you used the Chi-square test to compute (estimate) effect size (of what?): Chi-square tests are usually used to measure the strength of the association between categorical variables (e.g. contingency tables), expressed often in terms of p-values. The size of the effect (of what on what?) is not directly obtained from a chi-square test. Please add details

Results
-------
- The dream recall frequency: what do you mean by "recall" here?
- The difference between the “dog owner/dog(s) within the household” group and the “Never had a dog or lived with a dog” group was large (effect size = 1.079) and significant (see Table 4). Similarly, participants who had a dog or lived with a dog reported a higher dog dream percentage (effect size = 0.537) than participants that never lived with a dog (see Table 4). --> I couldn't find these estimated effect sizes in the Tables: where are these reported? The same holds for effect sizes mentioned henceforth.

Author Response

Responses to Reviewer 2

Dear authors,

your paper "Dreaming about dogs: An online survey" is generally well written and deals with a rather exotic yet not devoid of interest topic.
The absence of line numbers made the revision work more difficult: you may consider adding line numbers to the revised version of the paper. More importantly, essential details must ne added to Methods to improve the understanding of your results and the reproducibility of the study.

Specific comments

Introduction
------------
- of dog dreams significantly from 0.53% in 2006 (189 dreams) to 3.93% (280 dreams): I think that "in 2006" is a typo

Answer: The sentence was corrected, we meant in the year 2006 compared to the year 2007.

Methods
-------
- 2.1 Participants: where was the survey conducted?

Answer: Almost exclusive in Germany, Information was added.

- 2.3 Procedure: when you say ordinal regression, do you mean threshold models? (latent-variable model?) More details are needed on the model(s) used: please try to write the model explicitly, with the categorical (ordinal) variable to be analysed (response variable) and the independent variables used to model it. Otherwise it is not possible to fully understand your results and to try to reproduce your study

Answer: Information what variables are included was added. The exact sets of variables used for each regression analysis are depicted in the tables.

- 2.3 Procedure: it is not clear how you used the Chi-square test to compute (estimate) effect size (of what?): Chi-square tests are usually used to measure the strength of the association between categorical variables (e.g. contingency tables), expressed often in terms of p-values. The size of the effect (of what on what?) is not directly obtained from a chi-square test. Please add details

Answer: Information was added. The ordinal regression analysis in SAS uses Wald Chi-Squares for testing the effect of the single predictors (controlled for all other variables entered simultaneously in the analysis.

Results
-------
- The dream recall frequency: what do you mean by "recall" here?

Answer. Dream recall frequency is a fixed term, e.g., dream recall frequency is once per week. We reformulated the sentence for clarity.

- The difference between the “dog owner/dog(s) within the household” group and the “Never had a dog or lived with a dog” group was large (effect size = 1.079) and significant (see Table 4). Similarly, participants who had a dog or lived with a dog reported a higher dog dream percentage (effect size = 0.537) than participants that never lived with a dog (see Table 4). --> I couldn't find these estimated effect sizes in the Tables: where are these reported? The same holds for effect sizes mentioned henceforth.

Answer: The effect sizes are not reported in the Tables, only the significance levels. According to APA publication guidelines information should not be redundant (text and tables). We added the information that the significance levels are depicted in Table 4.

Reviewer 3 Report

This study appears to have been conducted during the COVID-19 pandemic. Although authors do compare with previous literature, one of the explanation why they had so many high recallers in their sample compared with previous samples might well be du to the fact that the pandemic created more sleep difficulties and hence more dream recall.  In the discussion, authors could develop more their thinking and discuss their results instead of merely repeating them. It would also have been interested to run some analyses on time spent with the animal vs active time spent with the animal according to age, which was not done. Authors, instead of focusing only on dogs could have also explored the place of all domestic animals in dreams (and life) to expand a bit more their horizons. Finally, authors self-referenced themselves a lot and should explorare other hypotheses and literature, and especially in the discussion. 

Author Response

Reviewer 3

This study appears to have been conducted during the COVID-19 pandemic. Although authors do compare with previous literature, one of the explanation why they had so many high recallers in their sample compared with previous samples might well be due to the fact that the pandemic created more sleep difficulties and hence more dream recall.

Answer: The Covid-19 pandemic issue was addressed, compared to previous studies carried out prior to Covid-19 dream recall frequency distribution was not different. This information was added..

In the discussion, authors could develop more their thinking and discuss their results instead of merely repeating them.

Answer: We faced the problem that we had no empirical studies to compare our findings to, so we could not add much information in the discussion (typically including these comparisons).

It would also have been interested to run some analyses on time spent with the animal vs active time spent with the animal according to age, which was not done.

Answer: I am not sure why the reviewer is interested in the relationship between time spent with the animal and age, for our purposes, we included age as a possible confounder to control for such effects if they are present. This might be interesting for a researcher interested in domestic animals, less so for a dream researcher and would – in our opinion – be beyond the scope of this paper.

Authors, instead of focusing only on dogs could have also explored the place of all domestic animals in dreams (and life) to expand a bit more their horizons.

Answer: Interesting suggestion for a future study. We started with dogs being the most popular domestic animal.

Finally, authors self-referenced themselves a lot and should explore other hypotheses and literature, and especially in the discussion. 

Answer: I am not sure what to do. We included all references we could find reporting empirical data on dog dreams. I am also not sure what other hypotheses we should explore, because the continuity hypothesis is the way to go, to look for the relationship between waking experiences with dogs and percentage of dreams with dogs and emotions of dog dreams. If the reviewer has specific suggestions, they are welcome.